# Estimation the Change in Liver Fibrosis Stage with Serial Measurement of Wisteria Floribunda Agglutinin-Positive Mac-2 Binding Protein in Metabolic Dysfunction-Associated Steatotic Liver Disease Patients

**DOI:** 10.3390/ijms26199410

**Published:** 2025-09-26

**Authors:** Tsuguru Hayashi, Yohei Kooka, Jo Kanazawa, Tomoki Matsuda

**Affiliations:** Department of Hepatology, Sendai Kousei Hospital, Aoba-ku, Sendai 980-0873, Japan; ykooka0416@gmail.com (Y.K.); jknzw5@gmail.com (J.K.); getomoki@gmail.com (T.M.)

**Keywords:** Wisteria floribunda agglutinin-positive Mac-2 binding protein, liver fibrosis, liver cirrhosis, noninvasive test

## Abstract

Assessment of liver fibrosis stage is important in the management of patients with metabolic dysfunction-associated steatotic liver disease (MASLD). However, non-invasive methods to determine the changes in liver fibrosis stage are unknown. We investigated whether Wisteria floribunda agglutinin-positive human Mac-2 binding protein (WFA^+^-M2BP), a serum fibrosis marker, can evaluate the changes in liver fibrosis stage. We observed the course of liver fibrosis stage and five serum fibrosis markers of 196 MASLD patients who had a paired biopsy performed. The changes in fibrosis markers, including WFA^+^-M2BP, were compared according to the changes in fibrosis stage. Factors associated with improvement of fibrosis stage were examined in a multivariate analysis. The changes in WFA^+^-M2BP (ΔWFA^+^-M2BP) were significantly correlated with changes in fibrosis stage (*p* < 0.01). The median of ΔWFA^+^-M2BP was −0.21, −0.08, −0.04, and 0.19 in 2 fibrosis stage regression group, 1 fibrosis stage regression group, fibrosis stage unchanged group, and fibrosis stage progression group, respectively (*p* < 0.01). However, other non-invasive markers did not reflect changes in fibrosis. ΔWFA^+^-M2BP was a significant factor for the regression of liver fibrosis stage in multivariate analysis (odds ratio: 3.54, 95% confidence interval: 1.55–8.12, *p* < 0.01). Time-course changes in WFA^+^-M2BP levels indicate the changes in liver fibrosis.

## 1. Introduction

Metabolic dysfunction-associated steatotic liver disease (MASLD) is a disease that afflicts about over 30 percent of world population [1,2]. In MASLD, the risk of liver-related and all-cause mortality increases with liver fibrosis progression [3,4,5]. Therefore, proper assessment of liver fibrosis stage is most important for the management of MASLD patients.

Liver biopsy is a gold standard for the evaluation of liver fibrosis. However, due to the invasiveness and sampling errors of liver biopsy, a variety of noninvasive methods to diagnosis liver fibrosis have been developed [6,7,8,9]. In MASLD, the Wisteria floribunda agglutinin-positive human Mac-2 binding protein (WFA^+^-M2BP) that was synthesized by hepatic stellate cells has a higher degree of diagnostic accuracy in predicting advanced liver fibrosis than non-invasive fibrosis markers, such as FIB-4 Index [10,11].

However, in these studies, liver fibrosis stage was diagnosed at only a single point. No study has evaluated the course of fibrosis stage and non-invasive fibrosis markers in the same MASLD patients. Therefore, it is unclear whether the change in non-invasive fibrosis markers could estimate the change in liver fibrosis stage. In fact, liver fibrosis stage progresses or regresses with lifestyle changes in MASLD [12,13]. However, it is impossible to judge progression or regression of liver fibrosis stage without liver biopsy.

In the present study, we investigated the fibrosis marker that could estimate the change in liver fibrosis stage.

## 2. Results

### 2.1. Patients Characteristics

Patients’ characteristics at the first biopsy and second biopsy were shown in Table 1. In the first biopsy, the mean age was 59 years, and 58.7% of patients were male. The period between the first and second biopsy was 2.2 years. In the second biopsy, aspartate aminotransferase, alanine aminotransferase, and gamma-glutamyltransferase levels decreased significantly (*p* < 0.01).

### 2.2. Fibrosis Stage and WFA^+^-M2BP in the First and Second Biopsy

In the first biopsy, the level of WFA^+^-M2BP was significantly elevated in accordance with progression of liver fibrosis stage (*p* < 0.01) (Figure 1A). The median WFA^+^-M2BP for stages 0, 1, 2, 3, and 4 were 0.50, 0.63, 0.74, 1.00, and 1.35, respectively. The optimal cut-off value of WFA^+^-M2BP to determine the advanced liver fibrosis stage was 0.76, with an AUROC of 0.76, a sensitivity of 79.5%, and a specificity of 65.1% (Figure 1B).

In the second liver biopsy, the level of WFA^+^-M2BP was significantly elevated in accordance with the progression of liver fibrosis (*p* < 0.01) (Figure 1C). The median WFA^+^-M2BP for stages 0, 1, 2, 3, and 4 were 0.51, 0.60, 0.85, 1.16, and 1.13, respectively. The optimal cut-off value of WFA^+^-M2BP to determine the advanced liver fibrosis stage was 0.82, with an AUROC of 0.88, a sensitivity of 89.7%, and a specificity of 76.6% (*p* < 0.01) (Figure 1D).

In WFA^+^-M2BP, the second liver biopsy showed a larger AUROC for predicting advanced liver fibrosis. In contrast, other fibrosis markers (FIB-4 Index, NFS, Type 4 Collagen and Hyaluronic acid) showed a smaller AUROC with the second liver biopsy. The first and second AUROC values for FIB-4 Index, NFS, Type 4 Collagen and Hyaluronic acid were 0.81 and 0.73, 0.82 and 0.77, 0.86 and 0.77, and 0.82 and 0.79, respectively.

### 2.3. Change in Fibrosis Stage and WFA^+^-M2BP

Of the 196 patients who was performed paired biopsy, 99 patients (50.5%) regressed fibrosis stage, 78 (39.8%) remained unchanged, and 19 (9.7%) progressed, respectively. In 99 patients who were had regressed fibrosis stages, 21 patients regressed two fibrosis stages. Change in fibrosis stage in all patients is shown in Table 2A.

The changes in WFA^+^-M2BP levels between first and second biopsy (ΔWFA^+^-M2BP) for each liver fibrosis stage are shown in Table 2B. When we divided all patients into four groups according to the change in fibrosis stage, ΔWFA^+^-M2BP was significantly correlated between the degree of change in the fibrosis stage (*p* < 0.01) (Figure 2). The median of ΔWFA^+^-M2BP was −0.21, −0.08, −0.04, and 0.19 in 2 fibrosis stage regression group, 1 fibrosis stage regression group, fibrosis stage unchanged group, and fibrosis progression group, respectively. The optimal cut-off value of ΔWFA^+^-M2BP to determine the regression of liver fibrosis stage was −0.24, with an AUROC of 0.65, a sensitivity of 32.3%, and a specificity of 89.7%.

The change in fibrosis stage was not associated with the level of WFA^+^-M2BP in the first liver biopsy. The median of levels of WFA^+^-M2BP in the first biopsy were 0.80, 0.70, 0.70, and 0.80 in 2 fibrosis stage regression group, 1 fibrosis stage regression group, fibrosis stage unchanged group, and fibrosis progression group, respectively (*p* = 0.24).

### 2.4. Change in Fibrosis Stage and Fibrosis Markers

FIB-4 Index, NFS, Type 4 Collagen and Hyaluronic acid did not correlate with the change in fibrosis stage (Table 3). Only WFA^+^-M2BP showed decrease in value with improvement of fibrosis stage (*p* < 0.01).

### 2.5. Factors Associated with Improvement of Liver Fibrosis

In univariate analysis, ΔWFA^+^-M2BP (odds ratio (OR): 4.16, 95% confidence interval (CI): 1.91–9.05, *p* < 0.01) and Δ nonalcoholic fatty liver disease activity score (NAS) (OR: 5.48, 95%CI: 2.96–10.20, *p* < 0.01) were factors associated with regression of liver fibrosis stage. Multivariate analysis demonstrated that ΔWFA^+^-M2BP (OR: 3.54, 95%CI: 1.55–8.12, *p* < 0.01) and ΔNAS (OR: 5.05, 95%CI: 2.69–9.51, *p* < 0.01) was an independent factor associated with improvement of fibrosis stage (Table 4).

Abbreviations: FIB-4 index, Fibrosis-4 index; NFS, non-alcoholic fatty liver disease fibrosis score; WFA^+^-M2BP, Wisteria floribunda agglutinin-positive human Mac-2 binding protein.

## 3. Discussion

To date, no fibrosis marker has previously been capable of reliably predicting changes in liver fibrosis. In this study, we demonstrated the impact of WFA^+^-M2BP on the prediction of changes in liver fibrosis stage. We have shown that the level of WFA^+^-M2BP decreased along the regression of the liver fibrosis stage, while that of WFA^+^-M2BP increased with the progression of the liver fibrosis stage. These results indicate that the measurement of time-course changes in WFA^+^-M2BP could be used to estimate the change in liver fibrosis stage. This fact is useful for the management of MASLD patients.

WFA^+^-M2BP is a serum glycan marker that is associated with liver fibrosis. The level of WFA^+^-M2BP was determined using a sandwich immunoassay with WFA and anti-M2BP antibody [14]. Previous studies have shown the utility of WFA^+^-M2BP for the estimation of advanced liver fibrosis [15,16,17]. WFA^+^-M2BP is a very useful noninvasive marker for chronic liver disease. Also, WFA^+^-M2BP could predict the development of hepatocellular carcinoma and liver related events [18,19,20,21]. The level of WFA^+^-M2BP is influenced by liver fibrosis and liver inflammation [22]. In this study, aspartate aminotransferase and alanine aminotransferase levels improved in the second biopsy. This could lead to an improvement in liver inflammation. Therefore, in this study, the diagnostic performance of advanced liver fibrosis with WFA^+^-M2BP was better in the second biopsy than in the first. Furthermore, ΔWFA^+^-M2BP was decreased in the fibrosis stage unchanged group. This indicates that WFA^+^-M2BP is a useful fibrosis marker when liver inflammation is resolved. These facts suggest that WFA^+^-M2BP could be an accurate indicator of changes in the stage of liver fibrosis.

Furthermore, WFA^+^-M2BP showed better diagnostic performance than FIB-4 Index, NFS, Type 4 Collagen, and Hyaluronic acid. This could be because WFA^+^-M2BP had higher diagnostic accuracy for liver fibrosis stage in the second biopsy. In addition, the AUROC for advanced fibrosis with WFA^+^-M2BP was larger than that of other fibrosis markers. The reason could be that in many cases, there was reduced liver inflammation at the second biopsy in our study cohort.

The strength of our study is that we performed two consecutive liver biopsies in a large number of cases. This could have led us to make direct comparisons. Liver biopsy is a useful method for assessing liver conditions in patients with chronic liver disease. However, in real-world practice, it is difficult to repeat a liver biopsy because of its invasiveness. Moreover, repeated liver biopsies every few years were not possible. Thus, this study is valuable because we clarified that the serial measurement of WFA^+^-M2BP is as meaningful as liver biopsy.

Our study has three limitations. First, this was a retrospective, observational study. Therefore, a prospective, randomized controlled study is required. Second, the number of liver biopsies in the same cases was two. Therefore, it may be more accurate to evaluate biopsies more frequently. Third, it could be better to extend the observation period between paired biopsies. However, further research is required.

We have shown that pathological findings could be identified by the course of WFA^+^-M2BP in patients with MASLD. This study is significant because MASLD is a chronic liver disease that requires a long-term follow-up. Furthermore, the incidence of liver-related events, cardiovascular events, and overall survival increases with the development of liver fibrosis. Therefore, an accurate assessment of the liver fibrosis stage with the measurement of WFA^+^-M2BP could allow for appropriate therapeutic intervention. MASLD is increasing all over the world, and proper management of MASLD is one of the social challenges [23,24,25]. Fibrosis markers have various roles not only in fibrosis diagnosis, but also in determining treatment efficacy and predicting carcinogenesis. However, it is unclear which fibrosis markers are useful for each purpose. WFA^+^-M2BP is a noninvasive test that can identify fibrosis changes.

In conclusion, we demonstrated that WFA^+^-M2BP could be useful for the real-time monitoring of liver fibrosis.

## 4. Materials and Methods

### 4.1. Patients Characteristics

This retrospective study enrolled 196 consecutive patients with MASLD who had repeated liver biopsies performed between February 2016 and June 2024 at Sendai Kousei hospital. All patients received dietary therapy and were instructed to restrict caloric intake between the first biopsy and second biopsy. WFA^+^-M2BP measurement, blood, and body weight were measured on the same day as the liver biopsy. WFA^+^-M2BP was measured in a blood test. The blood was stored at cooled temperature, and WFA^+^-M2BP was subsequently evaluated. Exclusion criteria were as follows: (1) infection with HBV or HCV, (2) the past presence of HCC, (3) alcohol abuse (alcohol intake of ≥30 g/day for men and ≥20 g/day for women). This study was approved by the ethics committees of Sendai Kousei Hospital and conformed to the ethical guidelines of the Declaration of Helsinki (approved number 7–20).

### 4.2. Assessment of Liver Fibrosis Stage

Ultrasound guided liver biopsy was performed using 16-gauge or 18-gauge needle. The liver biopsy specimen was confirmed to be at least 2 cm in size. Specimens were fixed, paraffin-embedded, and stained with hematoxylin-eosin and Masson’s trichrome. All liver biopsies were independently evaluated by two senior pathologists who were blinded to the clinical data. Liver fibrosis stage was defined by NASH CRN Histologic Scoring System [26].

### 4.3. Statistical Analysis

Categorial data were analyzed using the χ^2^ test or Fisher’s exact tests. Continuous variables were analyzed using the Mann–Whitney U test and expressed as median and interquartile range (IQR). *p* values < 0.05 were considered statistically significant. Changes in liver functions and fibrosis stage between first biopsy and second biopsy were analyzed by Wilcoxon single-rank test and χ^2^ test. Correlation between the level of WFA^+^-M2BP and fibrosis stage was analyzed by Kruskal–Wallis test. Receiver operating characteristic (ROC) curves and area under the ROC curve (AUROCs) were used to evaluate the diagnostic accuracy of liver fibrosis progression beyond stage 3. Factors associated with improvement in liver fibrosis were determined using logistic regression analysis. All variables with *p* value < 0.05 from the univariate analysis were included in the multivariate analysis. The cut off values that were used in logistic regression analysis were medians of all patients. The cut off value of ΔWFA^+^-M2BP was calculated by ROC curves. All statistical analyses were performed using the Statistical Package for the Social Science (SPSS) version 27 (SPSS Inc., Chicago, IL, USA) and Easy R (EZR) version 1.29 (Saitama Medical Center, Jichi Medical University, Saitama, Japan) [27].

## Figures and Tables

**Figure 1 ijms-26-09410-f001:**
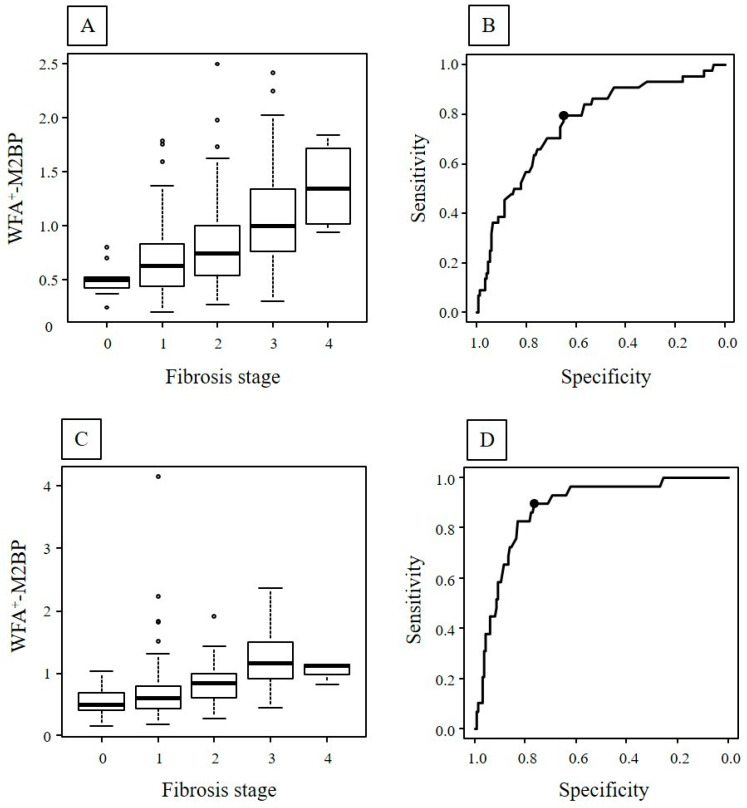
In the first liver biopsy, the level of WFA^+^-M2BP was elevated according to fibrosis stage (*p* < 0.01) (**A**). The boxplot represents the 25th to 75th percentiles and provides the interquartile range. The line through the box indicates the median value, and the error bar indicates the minimum and maximum non-extreme values. The area under the receiver operating characteristic curve was 0.76 (**B**). In the second liver biopsy, the level of WFA^+^-M2BP was elevated according to fibrosis stage (*p* < 0.01) (**C**). The area under the receiver operating characteristic curve was 0.88 (**D**).

**Figure 2 ijms-26-09410-f002:**
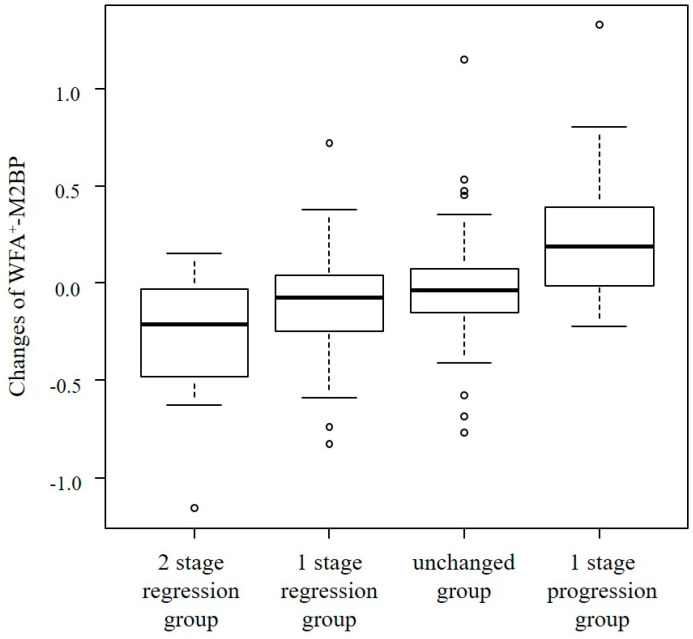
Correlation between the degree of change in liver fibrosis and ΔWFA^+^-M2BP. ΔWFA^+^-M2BP was significantly different between the four groups (*p* < 0.01).

**Table 1 ijms-26-09410-t001:** Patients’ characteristics at the first biopsy and second biopsy.

	First Biopsy	Second Biopsy	*p*-Value
Age (years)	59 (47–67)	61 (49–69)	<0.01
Gender (male/female)	115/81		
AST (U/L)	48 (34–73)	25 (21–34)	<0.01
ALT (U/L)	74 (53–117)	29 (21–50)	<0.01
GGT (U/L)	64 (43–90)	32 (22–51)	<0.01
TG (mg/dL)	133 (103–194)	118 (89–162)	<0.01
HbA1c	5.9 (5.6–6.5)	5.8 (5.5–6.1)	<0.01
Platelet counts (×10^9^/L)	23.3 (18.9–27.1)	23.1 (18.8–27.1)	0.23
Fibrosis stage: 0/1/2/3/4	10/85/57/38/6	41/105/21/26/3	<0.01
Body Weight (kg)	75.0 (64.6–87.1)	71.2 (59.9–82.8)	<0.01
BMI (kg/m^2^)	28.0 (25.8–31.0)	26.5 (23.9–29.8)	<0.01
WFA^+^-M2BP	0.72 (0.51–1.01)	0.65 (0.47–0.90)	<0.01
FIB-4 index	1.39 (0.93–2.23)	1.24 (0.84–1.77)	<0.01
NFS	−2.01 (−3.05–0.90)	−1.91 (−2.83–0.79)	<0.01
Type IV Collagen	118.5 (96.8–154.0)	110.0 (93.0–129.5)	<0.01
Hyaluronic acid	26.1 (12.2–46.8)	23.8 (12.2–43.4)	<0.01

Abbreviations: AST, aspartate aminotransferase; ALT, alanine aminotransferase; BMI, body mass index; FIB-4 index, Fibrosis-4 index; GGT, gamma-glutamyl transferase; HbA1c, hemoglobin A1c; NFS, non-alcoholic fatty liver disease fibrosis score; WFA^+^-M2BP, Wisteria floribunda agglutinin-positive human Mac-2 binding protein; TG, triglyceride.

**Table 2 ijms-26-09410-t002:** The changes in liver fibrosis stage between first and second biopsy. (**A**) Changes in Fibrosis stage. (**B**) Changes in WFA^+^-M2BP.

Fibrosis Stage at Second Biopsy						
Fibrosis Stage at First Biopsy	Stage 0	Stage 1	Stage 2	Stage 3	Stage 4	Total
(**A**)						
Stage 0	5	5				10
Stage 1	29	48	8			85
Stage 2	7	38	7	5		57
Stage 3		14	7	16	1	38
Stage 4				4	2	6
(**B**)						
Stage 0	−0.06	0.19				
Stage 1	−0.08	−0.03	0.21			
Stage 2	−0.21	0.10	−0.10	0.30		
Stage 3		−0.24	−0.07	0.03	0.14	
Stage 4				−0.18	−0.39	

Abbreviations: WFA^+^-M2BP, Wisteria floribunda agglutinin-positive human Mac-2 binding protein.

**Table 3 ijms-26-09410-t003:** The changes in noninvasive fibrosis markers between first and second biopsy.

	Changes in Fibrosis Stage				
	2 Stage Regression	1 Stage Regression	Unchanged	1 Stage Progression	*p*-Value
FIB-4 index	−0.37	−0.19	0.11	−0.14	0.31
NFS	0.11	0.16	0.21	0.13	0.88
WFA^+^-M2BP	−0.21	−0.08	−0.04	0.19	<0.01
Type IV Collagen	−29.00	−6.50	−10.00	10.00	<0.01
Hyaluronic acid	0.00	0.00	−1.05	0	0.60

Abbreviations: FIB-4 index, Fibrosis-4 index; NFS, non-alcoholic fatty liver disease fibrosis score; WFA^+^-M2BP, Wisteria floribunda agglutinin-positive human Mac-2 binding protein.

**Table 4 ijms-26-09410-t004:** Factor associated with improvement of fibrosis stage.

	Univariate Analysis			Multivariate Analysis		
	OR	95%CI	*p*-Value	OR	95%CI	*p*-Value
Age (<60 years)	0.75	0.43–1.31	0.31			
Sex (male)	0.91	0.52–1.61	0.75			
BMI (>30kg/m^2^)	0.61	0.33–1.11	0.11			
Advanced fibrosis (stage > 2)	1.39	0.71–2.73	0.34			
Reduction in body weight (>10%)	1.46	0.75–2.86	0.27			
ΔWFA^+^-M2BP (<−0.24)	4.16	1.91–9.05	<0.01	3.54	1.55–8.12	<0.01
ΔNAS (>2)	5.48	2.96–10.20	<0.01	5.05	2.69–9.51	<0.01

Abbreviations: FIB-4 index, Fibrosis-4 index; NFS, non-alcoholic fatty liver disease fibrosis score; WFA^+^-M2BP, Wisteria floribunda agglutinin-positive human Mac-2 binding protein.

## Data Availability

The dataset in this study is not publicly available due to patient privacy and ethical restrictions.

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
