# Peer review of "Estimation the Change in Liver Fibrosis Stage with Serial Measurement of Wisteria Floribunda Agglutinin-Positive Mac-2 Binding Protein in Metabolic Dysfunction-Associated Steatotic Liver Disease Patients"

_ijms, 2025, doi:10.3390/ijms26199410_

Round 1
Reviewer 1 Report
Comments and Suggestions for Authors
In this study, the different stages of liver fibrosis and the changes in expression of the Wisteria floribunda agglutinin-positive human Mac-2 binding protein (WFA+-M2BP) were evaluated in the serum of 196 patients with MASLD who underwent paired biopsy.
Here are my comments related to this manuscript.
-The authors should justify the study and clarify the new contribution of this study because it was reported in a previous published article that Wisteria floribunda agglutinin-positive mac-2 binding protein as an age-independent fibrosis marker in nonalcoholic fatty liver disease.
-Please update the prevalence of MASLD in the next sentence: “Metabolic dysfunction-associated steatotic liver disease (MASLD) is a disease that afflicts about 20 percent of world population. 1, 2”
-The exclusion criteria are unclear; for example, what comorbidities participants had and which ones were excluded.
-What size were the biopsy samples obtained from the participants?
-It also includes the criteria for the different stages of liver fibrosis considered in this study.
-Include in materials and methods how WFA+-M2BP was assessed in participants.
-Clarify whether serum samples were kept at -80°C and then evaluated.
-Clarify whether serum samples were kept at -80°C and then evaluated.
-What criteria were used to obtain the second biopsy in the participants?
-Please clarify whether the biopsies analysis for inflammation markers indicates agreement with the following sentence from the discussion section: “Furthermore, ΔWFA+-M2BP was decreased in the fibrosis stage unchanged group. This indicates that WFA+-M2BP is a useful fibrosis marker when liver inflammation is resolved.”
Comments on the Quality of English LanguageThere are some words with grammatical errors throughout the manuscript.
Author Response
Dear Editors and Reviewers
Thank you very much for taking your time and reviewing our manuscript. We are grateful for offering valuable advice. We have addressed your comments with point-by-point responses, and revised the manuscript accordingly. We believe that this manuscript has improved thanks to your advice. We strongly wish this manuscript to be published in International Journal of Molecular Sciences.
Reviewer 1
In this study, the different stages of liver fibrosis and the changes in expression of the Wisteria floribunda agglutinin-positive human Mac-2 binding protein (WFA+-M2BP) were evaluated in the serum of 196 patients with MASLD who underwent paired biopsy. Here are my comments related to this manuscript.
The authors should justify the study and clarify the new contribution of this study because it was reported in a previous published article that Wisteria floribunda agglutinin-positive mac-2 binding protein as an age-independent fibrosis marker in nonalcoholic fatty liver disease.
WFA⁺-M2BP is a marker of liver fibrosis in MASLD. However, no studies have clearly demonstrated whether changes in fibrosis markers correlated with changes in liver fibrosis stage in MASLD. In our study, WFA⁺-M2BP was the only fibrosis marker capable of predicting changes in liver fibrosis. We added in Discussion (Line 167).
Please update the prevalence of MASLD in the next sentence: “Metabolic dysfunction-associated steatotic liver disease (MASLD) is a disease that afflicts about 20 percent of world population. 1, 2”
We corrected (Line 33).
The exclusion criteria are unclear; for example, what comorbidities participants had and which ones were excluded.
Exclusion criteria were as follows: (1) infection with HBV or HCV, (2) the past presence of HCC, (3) alcohol abuse (intake of alcohol equivalent to 40 g/day or more). These are written in Methods (Line 58-60).
What size were the biopsy samples obtained from the participants?
Liver biopsy was performed using 16-gauge or 18-gauge needle. The liver biopsy specimen was confirmed to be at least 2cm in size. We added it (Line 64-65).
It also includes the criteria for the different stages of liver fibrosis considered in this study.
Liver fibrosis stage was defined by NASH CRN Histologic Scoring System. This is written in Line 68-69.
Include in materials and methods how WFA+-M2BP was assessed in participants.
WFA⁺-M2BP was measured in blood tests. We added it (Line 56-57).
Clarify whether serum samples were kept at -80°C and then evaluated.
The blood was stored at cooled temperature, and WFA⁺-M2BP was subsequently evaluated (Lone 57-58).
What criteria were used to obtain the second biopsy in the participants?
Patients who provided informed consent for a second liver biopsy were included in the study.
Please clarify whether the biopsies analysis for inflammation markers indicates agreement with the following sentence from the discussion section: “Furthermore, ΔWFA+-M2BP was decreased in the fibrosis stage unchanged group. This indicates that WFA+-M2BP is a useful fibrosis marker when liver inflammation is resolved.”
This study did not clarify whether the level of WFA⁺-M2BP are influenced by liver inflammation. However, this has been established in previous study (Reference 24).
Reviewer 2 Report
Comments and Suggestions for Authors
This study, which investigated the relationship between changes in liver fibrosis and fibrosis markers in steatotic liver disease through repeated biopsies in numerous cases, is considered both interesting and of significant clinical importance. However, there are several problems with publishing this paper, as pointed out below.
- ΔWFA⁺-M2BP was the only marker associated with fibrotic changes, yet the discussion regarding why only WFA⁺-M2BP was significant—rather than other markers such as FIB-4, NFS, type IV collagen, or hyaluronic acid—is insufficient. The authors should provide more detailed reference to the differences between WFA⁺-M2BP and the other markers used in this study, including past literature and molecular structural differences.
- Even when the change in fibrosis markers is the same, the progression of fibrosis stage may differ between groups with initially high and low marker levels. It should be demonstrated whether there is an association between the change in fibrosis and the level of fibrosis markers at the initial liver biopsy, or the mean values of fibrosis markers at the first and second biopsy.
- Weight changes and fibrosis markers should also be included in the background table. Weight changes may serve as a simpler indicator of fibrosis improvement.
- Regarding the definition of MASH, this study includes alcohol consumption equivalent to MetALD. It should be separately examined whether similar results occur in populations adhering to the MASH alcohol criteria (men <30g/day, women <20g/day). Alternatively, the study should explicitly state that the subjects comprise a population including both MASH and MetALD.
- The term “HR” shown in the results(3.5) should be changed to “OR.”
Author Response
Dear Editors and Reviewers
Thank you very much for taking your time and reviewing our manuscript. We are grateful for offering valuable advice. We have addressed your comments with point-by-point responses, and revised the manuscript accordingly. We believe that this manuscript has improved thanks to your advice. We strongly wish this manuscript to be published in International Journal of Molecular Sciences.
Reviewer 2
This study, which investigated the relationship between changes in liver fibrosis and fibrosis markers in steatotic liver disease through repeated biopsies in numerous cases, is considered both interesting and of significant clinical importance. However, there are several problems with publishing this paper, as pointed out below.
ΔWFA⁺-M2BP was the only marker associated with fibrotic changes, yet the discussion regarding why only WFA⁺-M2BP was significant rather than other markers such as FIB-4, NFS, type IV collagen, or hyaluronic acid is insufficient. The authors should provide more detailed reference to the differences between WFA⁺-M2BP and the other markers used in this study, including past literature and molecular structural differences.
This is because WFA⁺-M2BP had higher diagnostic accuracy for liver fibrosis stage in the second liver biopsy. Furthermore, in the second biopsy, the AUROC for advanced liver fibrosis with WFA⁺-M2BP was larger than that of other fibrosis markers. We added these things in Results. The reason could be many cases showed reduced liver inflammation at the second biopsy in our study cohort (Line 188-193).
Even when the change in fibrosis markers is the same, the progression of fibrosis stage may differ between groups with initially high and low marker levels. It should be demonstrated whether there is an association between the change in fibrosis and the level of fibrosis markers at the initial liver biopsy, or the mean values of fibrosis markers at the first and second biopsy.
The change of fibrosis stage was not associated with the levels of WFA⁺-M2BP in the first liver biopsy. We added in Results (Line 135-138).
Weight changes and fibrosis markers should also be included in the background table. Weight changes may serve as a simpler indicator of fibrosis improvement.
We added (Table 1).
Regarding the definition of MASH, this study includes alcohol consumption equivalent to MetALD. It should be separately examined whether similar results occur in populations adhering to the MASH alcohol criteria (men <30g/day, women <20g/day). Alternatively, the study should explicitly state that the subjects comprise a population including both MASH and MetALD.
All patients consumed alcohol below the specified MASLD diagnostic criteria (30g for men, 20g for women), and the description has been revised accordingly (Line 59-60).
The term “HR” shown in the results(3.5) should be changed to “OR.”
We corrected (Table 4).
Round 2
Reviewer 1 Report
Comments and Suggestions for Authors
I have no comments my suggestions were taken into account.
Author Response
Dear Editors and Reviewers
Thank you very much for taking your time and reviewing our manuscript. We sincerely thank the editors and reviewers for valuable comment and suggestion. We have addressed your comment with point-by-point response, and revised the manuscript accordingly. We believe the quality of this paper has been further improved.
Reviewer 2 Report
Comments and Suggestions for Authors
Thank you for submitting the revisions promptly.
The authors should revise the following sections.
Minor
1. Results 3.5. (lines 156-161) were all performed using logistic regression analysis and should use odds ratios instead of hazard ratios. Table 4 should also revert to OR.
Author Response
Dear Editors and Reviewers
We sincerely thank the editors and reviewers for valuable comment and suggestion. We have addressed your comment with point-by-point response, and revised the manuscript accordingly. We believe the quality of this paper has been further improved.
Thank you for submitting the revisions promptly.
The authors should revise the following sections.
Minor
1. Results 3.5. (lines 156-161) were all performed using logistic regression analysis and should use odds ratios instead of hazard ratios. Table 4 should also revert to OR.
Your comment is very important. We changed from HR to OR(Line 156-160).